# Impact of Subclinical *Haemoproteus columbae* Infection on Farmed Domestic Pigeons from Central Java (Yogyakarta), Indonesia, with Special Reference to Changes in the Hemogram

**DOI:** 10.3390/pathogens10040440

**Published:** 2021-04-07

**Authors:** Imron Rosyadi, Siti Isrina Oktavia Salasia, Bayanzul Argamjav, Hiroshi Sato

**Affiliations:** 1Laboratory of Parasitology, Joint Graduate School of Veterinary Medicine, Yamaguchi University, 1677-1 Yoshida, Yamaguchi 753-8511, Japan; imron.rosyadi@ugm.ac.id (I.R.); argamjav_bayanzul@muls.edu.mn (B.A.); 2Department of Clinical Pathology, Faculty of Veterinary Medicine, Gadjah Mada University, Bulaksumur, Yogyakarta 55281, Indonesia; isrinasalasia@ugm.ac.id

**Keywords:** haemosporidia, *Haemoproteus columbae*, domestic pigeon, *Columba livia* f. *domestica*, anemia, hemogram, *cytb* lineage, Indonesia

## Abstract

Pigeon haemoproteosis caused by *Haemoproteus columbae* (Apicomplexa: Haemosporida: Haemoproteidae) is globally prevalent in rock doves (*Columba livia*), although little is known regarding this disease in pigeons and doves in Indonesia. Blood samples of 35 farmed domestic pigeons (*C. livia* f. *domestica*) from four localities in Yogyakarta Special Region, Central Java, Indonesia, were collected from March to June, 2016, subjected to a hemogram, and analyzed for the presence of hemoprotozoan infections. Microscopic examination of blood smears revealed a prevalence of 62.5–100% of *H. columbae* at the four localities (*n* = 8–10 for each locality), and geometric means of 3.0–5.6% of erythrocytes were parasitized by young and mature gametocytes, suggesting that all infected pigeons were in the chronic phase of infection with repeated recurrences and/or reinfections. Nucleotide sequencing of mitochondrial cytochrome *b* gene (*cytb*) for haemosporidian species demonstrated the distribution of four major *cytb* lineages of *H. columbae* (mainly HAECOL1, accompanied by COLIV03, COQUI05, and CXNEA02 according to the MalAvi database). Hemogram analysis, involving the estimation of packed cell volume, erythrocyte counts, mean corpuscular volume, mean corpuscular hemoglobin concentration, and plasma protein and fibrinogen levels of 20 parasitized pigeons and five non-infected pigeons demonstrated significant macrocytic hypochromic anemia with hypoproteinemia and hyperfibrinogenemia in the infected pigeons. This study shows the profound impact of long-lasting subclinical pigeon haemoproteosis caused by *H. columbae* on the health of farmed domestic pigeons.

## 1. Introduction

Avian haemosporidian infection is caused mainly by the genera *Plasmodium* Marchiafava et Celli, 1885, *Haemoproteus* Kruse, 1890, or *Leucocytozoon* Sambon, 1908 (Apicomplexa: Haemosporidia). There are more than 200 nominal haemosporidian species and presumably thousands of undescribed cryptic species, which are differentiated genetically but often exhibit convergent morphology with other known taxa [1,2,3,4]. In rock doves (*Columba livia* Gmelin, 1789), including domestic pigeons (*C. livia* f. *domestica* Gmelin, 1789), *Haemoproteus columbae* Kruse, 1890 is globally the representative cause of the haemosporidian infection [2,5,6]. In addition to this species, at least five other *Haemoproteus* spp. infect pigeons and doves (Columbiformes: Columbidae), are transmitted by hippoboscid flies, and are classified in the subgenus *Haemoproteus* Kruse, 1890. In contrast, a large variety of birds from other families are infected by more than 126 different *Haemoproteus* spp. that are transmitted by biting midges (Ceratopogonidae) and are classified in the subgenus *Parahaemoproteus* Bennett, Garnham and Fallis, 1965 [2,7]. *Haemoproteus* spp. of these two subgenera are transmitted by different families of vectors, are phylogenetically distinct, and form separate clades [2,3,8,9,10,11,12,13].

Pigeon haemoproteosis caused by *H*. (*Haemoproteus*) *columbae* is transmitted by the bite of the pigeon louse fly *Pseudolynchia canariensis* (Macquart, 1839). It has a prepatent period of 22–38 days (ca. 30 days in average), wherein repeated merogony occurs in the lungs [2,14,15,16]. Merozoites released from the meront invade erythrocytes and develop into gametocytes. Pathogenicity of natural *H. columbae* infections is conventionally believed to be virtually minimal and of no veterinary importance [2,6,17]. However, in the accidental non-columbid hosts, such as captive naive birds translocated from non-endemic areas to zoos and aviaries in the endemic areas, clinical and moribund *Haemoproteus* (*Parahaemoproteus*) infections have been often reported [2,6,15]. In cases of accidental infection, severe haemoproteosis and death of infected birds have been found to be closely related to the pre-erythrocytic stage of parasites that occurs before the development of parasitemia. This illness is caused by the rupture of megalomeronts in the muscle and liver, and subsequent pathogenic changes [18,19,20,21,22,23]. In other words, virulence of *Haemoproteus* infection is seen in “abortive” infections. Thence, pathogenic mechanisms underlying accidental fatal or moribund cases caused by members of the subgenus *Parahaemoproteus* can be extended to explain the potential pathogenicity of *H.* (*Haemoproteus*) *columbae* infection in the natural host, although the latter species does not have megalomeronts in its life-cycle [24]. Recently, however, Nebel et al. [25] demonstrated a negative relationship between the level of *H. columbae* parasitemia and body mass in feral domestic pigeons in urban areas of Cape Town; a 1% increase in the number of infected erythrocytes was correlated with a 5.42 g reduction in body mass.

In the present study, we examined the prevalence and levels of parasitemia of haemosporidian infection in farmed domestic pigeons that are bred for racing and meat, generally called “squab” [26,27], at four localities in Yogyakarta Special Region, Central Java, Indonesia. Haemosporidians detected in the blood were genetically characterized based on mitochondrial DNA sequences of partial cytochrome *b* gene (*cytb*), as recommended by Bensch et al. [4]. To assess the impact of natural haemosporidian infection on the health of the pigeons, analyses of several hemogram parameters were performed.

## 2. Results

### 2.1. Prevalence and Morphology of Haemosporidian Species

Upon microscopic examination, 85.7% (30/35) of blood samples from domestic pigeons exhibited the presence of *Haemoproteus* sp. The geometric mean of infection intensity (parasitemia levels) was higher at Mlati (3.9% (*n* = 10)) and Ngemplak (5.6% (*n* = 9)), wherein all pigeons were infected with the haemosporidian species, and comparatively lower at Kalasan (3.9% (*n* = 6)) and Sedayu (3.0% (*n* = 5)), wherein two and three pigeons were uninfected, respectively (Table 1). We identified the species as *H. columbae* (Table 2 and Figure 1), based on the morphological features of the gametocytes [2], and we did not detect any other haemosporidian species. Mature gametocytes, which were sausage-shaped with blunt ends, were observed to displace the nucleus of host erythrocytes laterally. Parasitized erythrocytes were enlarged (Table 2), but not evidently deformed (Figure 1). Dimensions of non-parasitized erythrocytes in pigeons infected by *H. columbae* were significantly larger than those of uninfected pigeons (see footnote of Table 2). Macrogametocytes had a darker cytoplasm with randomly dispersed pigment granules than microgametocytes which had pigment granules with a polarized distribution. Nuclei were located in the central one-third area of gametocytes. In most cases, immature gametocytes were more frequently observed than mature gametocytes (Table 1). Usually a single erythrocyte contained one gametocyte, but occasionally they also had two gametocytes.

### 2.2. Molecular Characterization of Cytb Sequences, and Phylogenetic Analyses

Sixteen *cytb* nucleotide sequences from 14 pigeons were successfully obtained—seven 571 bp-long sequences and nine 478 bp-long sequences with six polymorphic sites. Basic Local Alignment Search Tool (BLAST) search confirmed that all sequences were partial *cytb* sequences of *H. columbae*. Consequently, four lineages (HAECOL1, COLIV03, COQUI05, and CXNEA02 according to MalAvi database [4]) were differentiated (Table 3), wherein there was one dominant lineage (HAECOL1; 11 sequences), and three other lineages (one or two sequences/lineage). All these lineages were translated into an identical amino acid sequence. Mixed infection of *H. columbae* of two lineages was found in the blood samples of two pigeons. A phylogenetic tree of *Haemoproteus* (*Haemoproteus*) spp. that infect pigeons and doves of Columbidae was constructed based on the *cytb* sequences (Appendix A).

### 2.3. Hematological Analyses

Twenty-five blood samples from 20 *H. columbae*-infected pigeons and five uninfected pigeons were subjected to a hemogram test (Table 4). Upon comparison of the infected and uninfected groups, statistically significant reductions were found in erythrocyte numbers, packed cell volume (PCV), mean corpuscular hemoglobin (MCH), mean corpuscular hemoglobin concentration (MCHC), hemoglobin concentration, and plasma protein concentration/mm^3^, whereas statistically significant increases were found in leukocyte numbers, mean corpuscular volume (MCV), and fibrinogen concentration/mm^3^. Increases in the leukocyte numbers were largely ascribed to increased numbers of monocytes. This hemogram profile suggests that *H. columbae* infection causes a statistically-significant degree of macrocytic hypochromic anemia with hypoproteinemia and inflammation.

Relationships between parasitemia level and degree of anemia (number of erythrocytes/mm^3^ and MCHC) are shown in Figure 2. There was a statistically significant linear negative relationship between parasitemia level and number of erythrocytes (1% increase in number of infected erythrocytes led to a 6 × 10^4^ decrease in erythrocytes/mm^3^; *R*^2^ = 0.297) or MCHC (1% increase in number of infected erythrocytes lead to a 0.49 g/dL decrease in MCHC; *R*^2^ = 0.285).

## 3. Discussion

In this study, we demonstrated a high prevalence (85.7%) of *H. columbae* infection in farmed domestic pigeons from Yogyakarta, Central Java; this is comparable to previous studies that have reported a *H. columbae* prevalence of 72.7% (24/33) or 96.9% (186/192) in feral domestic pigeons in Cape Town, South Africa [25,30], a prevalence of 82% (41/50) in the island of Tenerife, Canary Archipelago, Spain [31], a prevalence of 57.3% (59/103) in Qena, Egypt [32], and a prevalence of 100% (20/20) in São Paulo, Brazil [33].

Based on the high levels of prevalence and mean levels of parasitemia ranging from 3.0% to 5.6% of erythrocytes in the examined pigeons in this study and the temporal changes seen in parasitemia in domestic pigeons experimentally infected with *H. columbae* [16,34], it is likely that all or a majority of infected pigeons examined in our study might be in the long-lasting chronic phase of *H. columbae* parasitemia with evident recurrences or continued reinfections, which were demonstrated by erythrocytes actively parasitized not only with mature gametocytes, but also young gametocytes, as observed in the blood films (Table 1; Figure 1). In immunocompetent natural vertebrate hosts for *H. columbae*, i.e., *C. livia*, pre-erythrocytic growth or merogony occurs mainly in the lungs (the prepatent period ranging between 22 and 37 days), followed by gametogony in the erythrocytes after the cell invasion of merozoites (the acute phase ranging between nine and 20 days), and is ended by crisis [14,34]. Primary as well as recurrent infection induces a partial immunity to *H. columbae* reinfection, i.e., incomplete premunition which allows for superinfection [2,16,34,35], resulting in the long-lasting parasitemia with immature and mature gametocytes, as seen in this study. Here, the parasitemia levels (1.1–25.7% (geometric mean, 6.1%)) with high degrees of coexistence between immature and mature gametocytes (0.47–15.0 (geometric mean, 2.4):1, respectively) in the blood films of all 30 infected farmed pigeons suggested the existence of growing exoerythrocytic forms, such as merozoites, that might be continuously released from meronts in the lungs of pigeons bitten by pigeon louse flies infected with *H. columbae*.

As described above, in the natural avian hosts for *Haemoproteus* spp., the pathogenicity of the haemosporidians is conventionally believed to be virtually minimal, especially when compared with the pathogenicity of the genera *Plasmodium* and *Leucocytozoon* in avian hosts [2,6,17,34]. In accidental hosts, such as captive birds in zoos and aviaries; however, clinical and fatal *Haemoproteus* infections have been reported [6,18,19,21,22,23,33,36]. In such accidental hosts, severe haemoproteosis and death in infected birds have been reported mostly in the prepatent period due to the rupture of megalomeronts in muscle and liver and subsequent pathogenic changes [6].

Some evidence has shown that natural haemoproteosis can exert an important selective pressure on the survival, reproductive success, behavior, and community structure of the hosts, as evidenced by studies that involve the experimental interruption of natural *Haemoproteus* infection and analysis of its consequences [37,38,39,40,41]. Recently, Nebel et al. [25] demonstrated a negative relationship between *H. columbae* parasitemia level and body mass in feral domestic pigeons in urban areas of Cape Town. In this study, we demonstrated subclinical but substantial symptoms such as macrocytic hypochromic anemia and hypoproteinemia in *H. columbae*-infected pigeons, and we speculate that this could cause body mass reduction and other negative health conditions. The demonstration of these substantial negative effects on the health of domestic pigeons triggered by the latent infection by *H. columbae* is important as domestic pigeons are used as racing birds or for meat production. Cooked meats of farmed pigeons, such as “squab”, is a delicious, high-quality product that is consumed across many cultures or countries, such as European countries, U.S.A., Egypt, and Asian countries, including Indonesia [26,27,32,42]. Although negative correlation between parasitemia level and development of anemia was found to be statistically significant, the values of the coefficient determination was fairly low, less than 0.3 (Figure 2). As speculated above, the pigeons sampled in this study were chronically infected ones with reinfections. Therefore, a point parameter, i.e., levels of parasitemia, might not be the best in reflecting the actual state of infection. We suppose that an experimental observation on pigeons regularly exposed to *H. columbae*-carrying *P. canariensis* would clearly show the negative impact of natural *H. columbae* infection in pigeons and doves, in contrast to experimental *H. columbae* infections, wherein a single exposure to infected pigeon louse flies was used [43].

The MalAvi database is a unified taxonomic database for avian haemosporidian genera of *Plasmodium*, *Haemoproteus,* and *Leucocytozoon* [4], and the epidemiological or ecological significance of genetic lineages of isolated haemosporidian species can be understood with ease. The core sequence lengths of *cytb*, 478-bp for *Plasmodium* and *Haemoproteus* spp. and 479 bp-long for *Leucocytozoon* spp., cover almost all taxonomically important variations. For example, mitochondrial *cytb* lineages of parasites differing by as few as one nucleotide frequently indicate distinctly different areas of transmission and range of host species [44,45,46], although whether haemosporidians showing such closely similar *cytb* lineages represent independent species require detailed analyses of multiple genetic markers as well as other biological characteristics [4]. In the present study, we detected four *cytb* lineages (HAECOL1, COLIV03, COQUI05, and CXNEA02 according to the MalAvi database for *cytb* lineages) that are prevalent *H. columbae* lineages recorded worldwide, such as in Botswana, Cameroon, Nigeria, South Africa, Brazil, Colombia, and Italy [11,25,33,44,47,48].

The ecology and biting behavior of hippoboscid flies is substantially different from those of biting midges and might explain the highest host specificity of *H. columbae* to pigeons and doves. Adult stages of pigeon louse flies are virtually flightless and crawl on the host body surface [7]. Detecting only *H. columbae* in farmed domestic pigeons in Yogyakarta, Central Java, suggests feasible control of the infection by vector control and better housing. This preventative measure has advantages when considering the findings of the present study that *H. columbae* infection causes subclinical but substantial hypochromic anemia and hypoproteinemia.

## 4. Materials and Methods

### 4.1. Blood Collection and Microscopic Examination

Thirty-five healthy-looking domestic pigeons were purchased between March and June 2016 from local breeders at four localities in Yogyakarta Special Region, namely, Mlati (*n* = 10), Ngemplak (*n* = 9), Kalasan (*n* = 8), and Sedayu (*n* = 8), with at least 11–18 km distance between different localities. Pigeons were 6 to 18 months old; however, information such as sex and body weight were not recorded. Approximately 0.5–1.0 mL of blood was individually collected from the brachial vein using clean syringes with fine needles, and immediately transferred to 3-mL BD Vacutainer™ K_3_EDTA glass tubes (Fisher Scientific, Arendalsvägen, Göteborg, Sweden). Two to three thin blood films were prepared on clean glass slides for each blood sample, air-dried, and fixed in absolute methanol for 10 min. Blood films were stained with Giemsa’s solution (Sigma-Aldrich, St. Louis, MS, USA). Several drops of blood from 14 arbitrarily selected pigeons (Mlati (*n* = 4), Ngemplak (*n* = 5), Kalasan (*n* = 3), and Sedayu (*n* = 2)) were placed on a circle of Whatman™ FTA™ Classic Card (GE Healthcare UK Ltd., Amersham, Buckinghamshire, UK). The residual amount of blood was used for hematological analyses, which are mentioned below. After blood collection, all pigeons were released. All animal experiments were performed according to the Guidelines on Animal Experimentation as set out by the Institutional Animal Care and Use Committee (IACUC), Faculty of Veterinary Medicine, Gadjah Mada University.

Microscopic observation of blood films was first conducted at ×400 magnification, and then at ×1000 with oil immersion using an Olympus BX60 light microscope equipped with a DP72 digital camera (Olympus, Nishi-Shinjuku, Tokyo, Japan). The level of parasitemia was determined by counting the number of parasitized cells per 1000 erythrocytes, according to Godfrey et al. [49]. Measurements of parasites and their identification were conducted according to Valkiūnas [2]. Representative blood films collected in the present work were deposited in the National Museum of Nature and Science, Tokyo, Japan.

### 4.2. Hematological Analyses

Hematological analyses were manually conducted in the Department of Clinical Pathology, Faculty of Veterinary Medicine, Gadjah Mada University, referring to Benjamin [50] and Coles [51] with slight modifications. Parameters examined included number of erythrocytes, number of leukocytes, PCV, MCV, MCH, MCHC, plasma protein, and fibrinogen per mm^3^.

Three microhematocrit tubes, coated by heparin, were filled with pigeon blood in the BD Vacutainer™ K_3_EDTA glass tubes (Fisher Scientific, Arendalsvägen, Göteborg, Sweden), and centrifuged at 3500 rpm for 10 min. PCV value were determined using a microhematocrit tube reader. Plasma fraction of one tube was individually dispensed onto the prism of the Goldberg TS Meter refractometer (Reichert Technologies, Depew, NY, USA) to obtain total plasma protein concentration. Fibrinogen concentration was determined by the heat precipitation method. Briefly, the second tube was incubated in the 58 °C water bath for 3 min to precipitate the fibrinogen as a white ring around the bottom of plasma fraction. These microhematocrit tubes were recentrifuged, and plasma fraction was measured by the method described above to obtain plasma protein concentration without fibrinogen. Fibrinogen concentration was calculated as the difference of plasma protein concentrations of the first and second microhematocrit tubes. Numbers of erythrocytes and leucocytes were determined by the hemocytometer method using phosphate-buffered saline and Turk’s solution. Hemoglobin concentration was determined by the colorimetric method using Drabkin’s reagent (Sigma-Aldrich, St. Louis, MS, USA) following manufacturer’ instruction. MCV, MCH, and MCHC were calculated using the standard formulae.

### 4.3. DNA Extraction, Amplification and Sequencing

As mentioned above, blood from 14 arbitrarily selected pigeons (Mlati (*n* = 4), Ngemplak (*n* = 5), Kalasan (*n* = 3), and Sedayu (*n* = 2)) were used for DNA extraction. Using a 2-mm Harris Uni-Core puncher (Whatman^®^; GE Healthcare UK Ltd., Amersham, Buckinghamshire, UK), four punches from each sample circle of Whatman™ FTA™ Classic Card were placed in an Eppendorf tube. The DNA of each sample was extracted from these four punches using an Illustra™ tissue and cells genomicPrep Mini Spin Kit (GE Healthcare UK Ltd., Amersham, Buckinghamshire, UK) according to the instructions of the manufacturer. PCR amplification of mitochondrial *cytb* DNA fragments was performed in a 20-µL PCR solution containing DNA polymerase packed in Blend Taq-Plus- (TOYOBO, Dojima Hama, Osaka, Japan), a primer pair of the forward HaemNF1 (5′-CAT ATA TTA AGA GAA NTA TGG AG-3′) and the reverse HaemNR3 (5′-ATA GAA AGA TAA GAA ATA CCA TTC-3′) for haemosporidians of genera *Haemoproteus*, *Plasmodium*, and *Leucocytozoon* [52], and one µL of template DNA. Another primer pair of the forward HaemF (5′-ATG GTG CTT TCG ATA TAT GCA TG-3′) and the reverse HaemR2 (5′-GCA TTA TCT GGA TGT GAT AAT GGT-3′), usually used as a second-round primer pair of nested PCR using products of the aforementioned PCR [52], was independently performed [53]. The following PCR cycling protocol was used: 3 min at 94 °C followed by 35 cycles of 30 s at 94 °C, 30 s at 50 °C, and 60 s at 72 °C; this was followed by a final extension at 72 °C for 7 min. The PCR products were purified using a FastGene Gel/PCR Extraction Kit (NIPPON Genetics Co., Tokyo, Japan) and sequenced directly from both ends with the original primers. When direct sequencing was not satisfactory, the purified PCR products were cloned into the plasmid vector pTA2 (TArget Clone™; TOYOBO, Dojima Hama, Osaka, Japan) and transformed into *Escherichia coli* JM109 cells (TOYOBO, Dojima Hama, Osaka, Japan) according to the instructions of the manufacturer. Following propagation, the plasmid DNA was extracted using a FastGene Plasmid Mini Kit (NIPPON Genetics Co., Ltd., Bunkyoku, Tokyo, Japan) and inserts from multiple independent clones, at least three, were sequenced using universal M13 forward and reverse primers according to the instructions of the manufacturer (TOYOBO, Dojima Hama, Osaka, Japan). The nucleotide sequences obtained in the present study are available from the DDBJ/EMBL/GenBank databases under the accession nos. LC605998–LC606013.

### 4.4. Phylogenetic Analysis

Fragments of the newly obtained *cytb* sequences (571-bp or 478-bp) were analyzed to identify highly similar nucleotide sequences using the BLAST of the National Center for Biotechnology Information website. The lineage names were identified by performing BLAST searches in the MalAvi database (http://130.235.244.92/Malav i/; accessed on 26 January 2021) [4]. For phylogenetic analysis, the newly obtained *cytb* sequences and closely related sequences of *Haemoproteus* (*Haemoproteus*) spp. retrieved from the DDBJ/EMBL/GenBank databases were aligned using the CLUSTAL W multiple alignment program [54], with subsequent manual adjustments. Trimmed sequences of 465 characters, of which 99 were variable, were subjected to subsequent analysis. Maximum likelihood analysis was performed with the program, PhyML [55,56], provided on the “phylogeny.fr” website (http://www.phylogeny.fr/; accessed on 26 January 2021). The probability of inferred branching was assessed by the approximate likelihood-ratio test, an alternative to the nonparametric bootstrap estimation of branch support [57].

### 4.5. Statistical Analysis

Hemogram data obtained in the present study are expressed as a range with its mean or mean ± standard deviation in parentheses. Hemogram comparison between uninfected (*n* = 5) and infected pigeons (*n* = 20) were statistically analyzed using one-way analysis of variance, ANOVA (StatView ver. 5; Abacus Concepts Inc., Berkeley, CA, USA). Correlations between values of infection intensities and hemogram parameters were analyzed via regression analysis using the same statistical package. A *p*-value less than 0.05 was considered statistically significant.

## 5. Conclusions

Pigeon haemoproteosis caused by *H*. (*H.*) *columbae* is prevalent mostly in the tropics and subtropics, and the pathogenicity of natural *H. columbae* infection is conventionally believed to be virtually minimal and of no veterinary importance. However, we observed that farmed domestic pigeons in Yogyakarta, Central Java, Indonesia have an 85.7% (30/35) prevalence and a high level of parasitemia, wherein 3.0–5.6% erythrocytes were parasitized with immature and mature gametocytes, indicating a long-lasting subclinical infection with incomplete premunition allowing reinfection. Careful hemogram examination of pigeons with/without *H. columbae* parasitemia suggested that *H. columbae* infection might cause a statistically significant degree of macrocytic hypochromic anemia with hypoproteinemia and inflammation.

## Figures and Tables

**Figure 1 pathogens-10-00440-f001:**
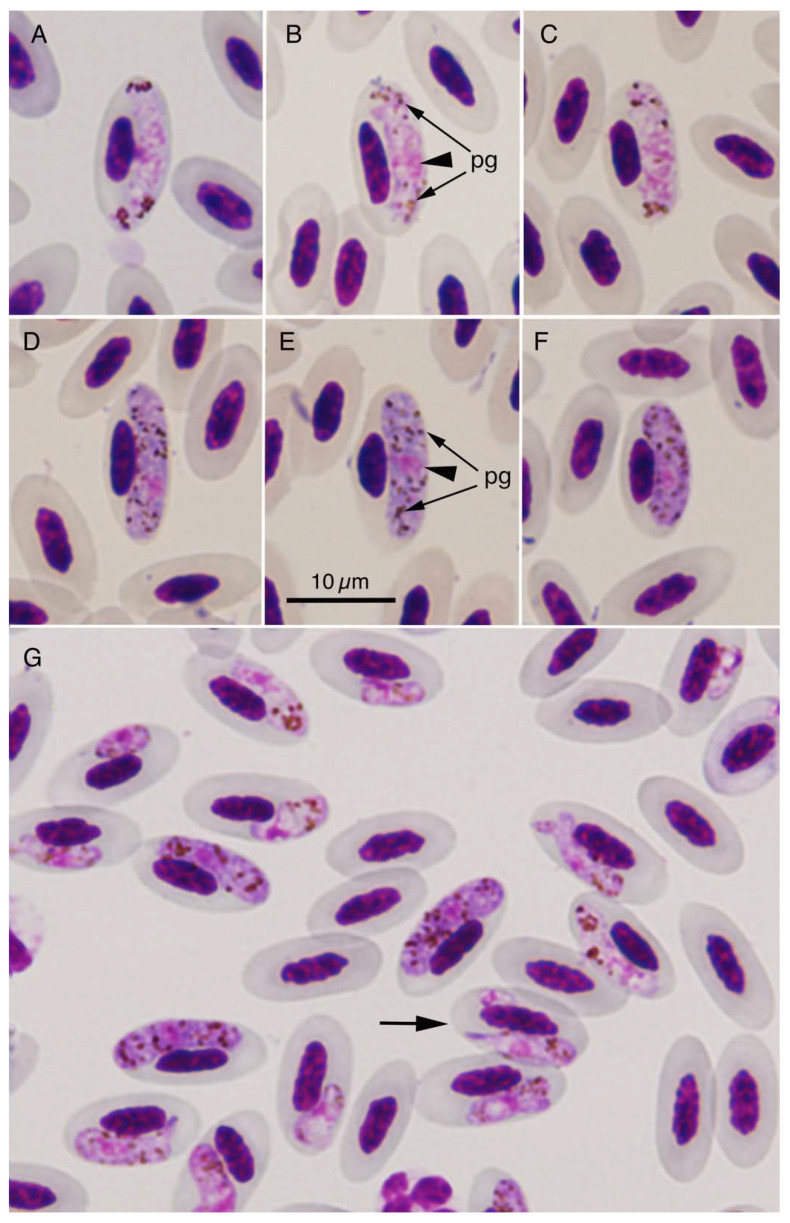
*Haemoproteus columbae* gametocytes in the blood of farmed domestic pigeons (*Columba livia* f. *domestica*) from Yogyakarta Special Region, Indonesia. Mature microgametocytes (**A**–**C**), mature macrogametocytes (**D**–**F**), and various developmental stages of gametocytes (**G**) in erythrocytes of a pigeon from Ngemplak, Yogyakarta. All photographs are at the same magnification, and the scale bar (10 µm) is shown in the photograph (**E**). Nuclei (arrowheads) and pigment granules (pg) of *H. columbae* are shown in the photographs (**B**,**E**). Arrow in the photograph (**G**) indicates an erythrocyte with two gametocytes.

**Figure 2 pathogens-10-00440-f002:**
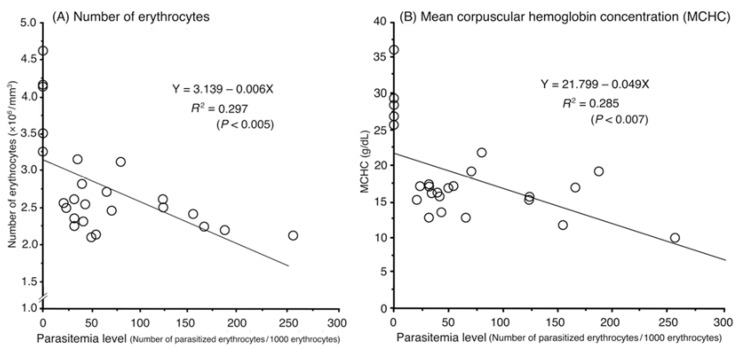
Relationship between *Haemoproteus columbae* parasitemia level (number of parasitized erythrocytes/1000 erythrocytes) and number of erythrocytes (×10^6^/mm^3^) (**A**) or mean corpuscular hemoglobin concentration (MCHC; g/dL) (**B**). Each open circle represents one individual farmed pigeon. Linear regressions shown in A and B are statistically significant with *R*^2^ = 0.297 and 0.285, respectively.

**Table 1 pathogens-10-00440-t001:** Farmed domestic pigeons examined with detected haemosporidians.

Locality	Number of Pigeons Examined	Prevalence of Haemosporidians	Detected Haemosporidians	Parasitemia (%) ^1^	Ratio of Immature Gametocytes: Mature Gametocytes
Mlati, Yogyakarta (7°43′53″ S, 110°19′52″ E)	10	100% (10/10)	*Haemoproteus columbae*	1.1–18.9 (3.9)	0.67–4.39 (1.61): 1
Ngemplak, Yogyakarta (7°41′57″ S, 110°26′42″ E)	9	100% (9/9)	*Haemoproteus columbae*	1.6–25.7 (5.6)	0.80–15.00 (3.35): 1
Kalasan, Yogyakarta (7°45′18″ S, 110°29′06″)	8	75.0% (6/8)	*Haemoproteus columbae*	2.5–6.7 (3.9)	2.15–11.75 (4.10): 1
Sedayu, Yogyakarta (7°48′49″ S, 110°16′17″)	8	62.5% (5/8)	*Haemoproteus columbae*	2.1–7.1 (3.0)	0.47–15.00 (1.47): 1

^1^ Percentage of parasitized cells by counting 1000 erythrocytes. Values are expressed by range with geometric mean in parentheses.

**Table 2 pathogens-10-00440-t002:** Morphometric values of erythrocytes and *Haemoproteus columbae* gametocytes in infected pigeons (expressed in µm).

Feature		The Present Study (*n* = 30)	Valkiūnas (2005) ^1^ (*n* = 31)
Uninfected erythrocyte ^2^			
	Length	12.3–14.9 (13.4)	12.8–14.7 (13.7)
	Width	6.6–7.7 (7.0)	6.4–7.7 (7.0)
	Length of nucleus	6.1–7.7 (6.7)	6.2–7.7 (6.7)
	Width of nucleus	2.3–3.4 (2.8)	2.1–2.9 (2.4)
Infected erythrocyte with a mature microgametocyte			
	Length	12.2–16.6 (14.1)	12.9–15.9 (14.4)
	Width	6.1–7.8 (7.0)	5.3–7.8 (6.9)
	Length of nucleus	5.5–7.4 (6.4)	5.5–7.4 (6.5)
	Width of nucleus	2.3–3.5 (2.7)	2.1–2.6 (2.3)
Infected erythrocyte with a mature macrogametocyte			
	Length	13.0–15.9 (14.5)	13.8–16.0 (15.0)
	Width	6.2–7.9 (7.2)	6.0–7.9 (7.1)
	Length of nucleus	4.9–7.5 (6.3)	6.0–7.4 (6.5)
	Width of nucleus	2.2–2.9 (2.6)	1.7–2.7 (2.3)
Mature microgametocyte			
	Length	10.5–16.6 (13.7)	11.6–15.5 (13.3)
	Width	2.7–5.0 (3.6)	2.6–4.3 (3.6)
	Length of nucleus	— ^3^	—
	Width of nucleus	2.6–4.5 (3.1)	2.6–4.3 (3.6)
Mature macrogametocyte			
	Length	12.4–16.4 (14.6)	13.4–16.7 (14.8)
	Width	2.5–4.2 (3.5)	3.0–4.2 (3.4)
	Length of nucleus	2.1–3.2 (2.6)	2.1–3.6 (2.9)
	Width of nucleus	1.5–3.2 (2.3)	1.5–3.4 (2.3)

^1^ According to Valkiūnas and Iezhova [28]. ^2^ Measurements of 10 erythrocytes each from five uninfected pigeons were as follows (*n* = 50 in total): Length, 9.5–12.2 (11.4); width, 4.6–7.8 (6.2); length of nucleus, 4.13–6.7 (5.6); and width of nucleus, 1.9–3.7 (2.7). Differences in all parameters of dimensions of uninfected erythrocytes, except for width of nucleus, between uninfected and *H. columbae*-infected pigeons were statistically significant (*p* < 0.05). ^3^ Difficult to measure due to unclear borders.

**Table 3 pathogens-10-00440-t003:** Detected *cytb* lineages of *Haemoproteus columbae* in farmed pigeons in Central Java.

Locality	Number of Pigeons Examined Molecular–Genetically	Detected *cytb* Lineages of *H. columbae* (Number of Pigeons)	DDBJ/EMBL/GenBank Accession No.
Mlati, Yogyakarta	4	HAECOL1 (3)	LC605998–LC606001
		COLIV03 (1)	
Ngemplak, Yogyakarta	5	HAECOL1 (5)	LC606002–LC606008
		CXNEA02 (2) ^1^	
Kalasan, Yogyakarta	3	HAECOL1 (2)	LC606009–LC606011
		COLIV03 (1)	
Sedayu, Yogyakarta	2	HAECOL1 (1)	LC606012–LC606013
		COQUI05 (1)	

^1^ Two pigeons were co-infected with two lineages (HAECOL1 and CXNEA02) of *H. columbae*.

**Table 4 pathogens-10-00440-t004:** Hemograms of pigeons infected and uninfected with *Haemoproteus columbae* gametocytes.

Parameter	Unit	Infected Pigeons (*n* = 20)	Uninfected Pigeons (*n* = 5)	Normal Range Reported by Ihedioha et al. [29] ^1^	Change ^2^	Statistical Significance
Number of erythrocytes	10^6^/mm^3^	2.10–3.15	3.25–4.60	2.12–3.95	↓	*p* < 0.001
		(2.49 ± 0.29)	(3.93 ± 0.55)	(3.34 ± 0.38)		
Number of leukocytes	10^3^/mm^3^	11.95–12.85	8.15–11.20	12.50–35.50	↑	*p* < 0.007
		(12.44 ± 0.34)	(9.61 ± 1.22)	(23.36 ± 7.06)		
PCV	%	34–50	45–48	32–55	↓	*p* < 0.001
		(39.8 ± 4.6)	(46.2 ± 1.6)	(44.5 ± 4.7)		
MCV	fl	134.10–218.60	97.83–138.46	109.82–169.09	↑	*p* < 0.001
		(161.04 ± 18.44)	(119.24 ± 15.34)	(133.86 ± 19.37)		
Hemoglobin	g/dL	4.5–10.9	11.5–17.2	7.76–16.00	↓	*p* < 0.001
		(6.30 ± 1.28)	(13.54 ± 2.26)	(12.89 ± 1.55)		
MCH	pg	18.60–35.16	26.30–41.45	n.d. ^3^	↓	*p* < 0.001
		(25.39 ± 4.11)	(34.79 ± 5.95)			
MCHC	g/dL	10.00–21.80	25.56–35.83	23.57–33.75	↓	*p* < 0.001
		(15.90 ± 2.72)	(29.22 ± 3.97)	(28.97 ± 2.59)		
Plasma protein	g/dL	1.0–4.0	4.1–5.2	n.d. ^3^	↓	*p* < 0.001
		(2.4 ± 1.0)	(4.7 ± 0.5)			
Fibrinogen	g/dL	0.2–1.0	0.1–0.4	n.d. ^3^	↑	*p* < 0.001
		(0.8 ± 0.3)	(0.2 ± 0.1)			

^1^ Values were calculated based on 64 adult domestic pigeons of both sexes. ^2^ Status of infected pigeons, compared with that of uninfected pigeons. ^3^ No data.

## Data Availability

Representative blood films collected in the present work were deposited in the National Museum of Nature and Science, Tokyo, Japan. The nucleotide sequences obtained in the present study are available from the DDBJ/EMBL/GenBank databases under the accession nos. LC605998–LC606013.

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
