# Peer review of "Impact of Subclinical Haemoproteus columbae Infection on Farmed Domestic Pigeons from Central Java (Yogyakarta), Indonesia, with Special Reference to Changes in the Hemogram"

_pathogens, 2021, doi:10.3390/pathogens10040440_

Round 1
Reviewer 1 Report
The reviewed manuscript presents interesting study on single host-parasite system of Haemoproteus columbae infecting feral pigeons (Columba livia forma domestica). The authors preformed sampling of purchased from the market feral pigeons in four locations in Yogyakarta, Indonesia and found quite high prevalence (over 80%) of haemoproteid infections among 35 pigeons. The main value of the manuscript is that it implements morphological and molecular identification of haemosporidian parasites as well as obtained parasitaemia values, which were compared with haematological parameters of the infected and uninfected pigeons. According the results (Fig. 3, Table 3) it appears that haematological parameters are strongly correlated with prevalence and parasitaemia of H. columbae. For me it is surprising to find such a strong correlation in chronically infected birds if we consider Haemoproteus infections relatively benign for the avian host. Unfortunately, there are no data on sex, age and weight as well as other morphometries of the pigeons representing their fitness, which might influence the results. The manuscript have several terminological and methodological issues that will require through revision or argumentation. Generally, I disagree with using of term “malaria” for infections by Haemorpteus spp. The term malaria should be attributed to Plasmodium spp only! Please check the following paper by ValkiÅ«nas et al. 2005 (doi: 10.1016/j.pt.2005.06.001) as argumentation.
My major concerns on the manuscript are mostly about methodological section.
First, hematological analysis are not well explained and have to be improved. It must be clear how were obtained the values for number of erythrocytes, number of leukocytes, PCV, MCV, MCH, MCHC, plasma protein, and fibrinogen per mm3. Please explain it in details with particular methods that were used for each value.
Second, for me the molecular methods for diagnosis and identification of haemosporidian parasites are not well explained and need improvement. It might be a grammar issue but for example on L291 it is not clear if Blend Taq-Plus is a sort of master mix or it was added buffers and dNTPs in the solution. The total volume of the original protocol is 25 μl but in the methods is mention 20 μl. Is that a total amount of solution or it is before adding the sample and primers? Please explain that in details! Furthermore, I am not sure if the authors performed nested PCR protocol described by Hellgren et al. 2004 correctly (L295-297). It is not clear how much of the sample they use for the first PCR. Did DNA concentration was quantify after extraction? What is authors’ meaning with "independently performed" second primer pair? According the protocol developed by Hellgre et al. 2004 it have to be use 1 microliter of the first PCR product to run the second PCR in order to increase the sensitivity of the method. As approval, it seems that the authors manage to obtain only 16 sequences out of 30 infected pigeon (L128-129).
Third, it is not clear why cloning procedure was applied to some of the samples (L302-307). There are also missing references for the universal primers M13. Cloning is usually applied for sequencing of co-infections in the same individual host (see Pérez-Tris & Bensch, 2005; doi:10.1017/S003118200500733X). Actually, it is not clear how the authors identify co-infections if there are any?
The performed phylogenetic analyses (Fig.2) did not bring new information and can be desecrated. It is also not clear how many cytb sequences are included in the phylogeny and why there is no outgroup to root the tree? In my opinion including identical sequences are not giving good results and can be excluded from the tree. Much representative perhaps would be to have a table with cytb lineages that were found, morphospecies, parasitaemia and location. As there is no taxonomical or systematical question in the manuscript, I found the phylogeny quite unnecessary.
The discussion and conclusions must be entirely revised. In its current version the discussion is too generous and not focused on the main results of the study, i.e. hematogram comparison with infection. Suggestions for superinfection based on the immature to mature gametocytes ratios is too speculative and vague. As the authors did not control the infection in experimental conditions it is hard to say if the young gametocytes infecting erythrocytes are due to reinfection by vector bite or recurrences of the meronts if the hard tissues of the host. I think it will be much more interesting to discuss the results in terms physiological and immunological manner as well.
Some of the same and other comments and suggestions I implemented in the attached PDF file.

Author Response
Thank you a lot for your kind review with invaluable comments and suggestions. Based on your comments, we have revised the manuscript.
A point-by-point responses to your comments are as follows;
The reviewed manuscript presents interesting study on single host-parasite system of Haemoproteus columbae infecting feral pigeons (Columba livia forma domestica). The authors preformed sampling of purchased from the market feral pigeons in four locations in Yogyakarta, Indonesia and found quite high prevalence (over 80%) of haemoproteid infections among 35 pigeons. The main value of the manuscript is that it implements morphological and molecular identification of haemosporidian parasites as well as obtained parasitaemia values, which were compared with haematological parameters of the infected and uninfected pigeons. According the results (Fig. 3, Table 3) it appears that haematological parameters are strongly correlated with prevalence and parasitaemia of H. columbae. For me it is surprising to find such a strong correlation in chronically infected birds if we consider Haemoproteus infections relatively benign for the avian host. Unfortunately, there are no data on sex, age and weight as well as other morphometries of the pigeons representing their fitness, which might influence the results. The manuscript have several terminological and methodological issues that will require through revision or argumentation. Generally, I disagree with using of term “malaria” for infections by Haemorpteus spp. The term malaria should be attributed to Plasmodium spp only! Please check the following paper by ValkiÅ«nas et al. 2005 (doi: 10.1016/j.pt.2005.06.001) as argumentation.
>>>Thank you for your understanding of the highlighted issues and importance of this study. Yes, we had overlooked collection of some basic and important data related to host fitness. We examined farmed pigeons, not feral pigeons.
Concerning to the issue of use of the term ‘malaria’, we had referred to the argument of Martinsen, Perkins and Schall (2008), who analyzed a three genome phylogeny of Plasmodium and closely related genera (Hepatocystis and Haemoproteus) [Reference no. 3 in this manuscript]. Finally, we agreed with their opinion to use the term ‘malaria’ not only for the disease caused by Plasmodium spp., but also those caused by Hepatocystis and Haemoproteus spp.
My major concerns on the manuscript are mostly about methodological section.
First, hematological analysis are not well explained and have to be improved. It must be clear how were obtained the values for number of erythrocytes, number of leukocytes, PCV, MCV, MCH, MCHC, plasma protein, and fibrinogen per mm3. Please explain it in details with particular methods that were used for each value.
>>>Thank you for your comments. We have added detail information related to the methodology of hematological analyses.
Second, for me the molecular methods for diagnosis and identification of haemosporidian parasites are not well explained and need improvement. It might be a grammar issue but for example on L291 it is not clear if Blend Taq-Plus is a sort of master mix or it was added buffers and dNTPs in the solution. The total volume of the original protocol is 25 μl but in the methods is mention 20 μl. Is that a total amount of solution or it is before adding the sample and primers? Please explain that in details! Furthermore, I am not sure if the authors performed nested PCR protocol described by Hellgren et al. 2004 correctly (L295-297). It is not clear how much of the sample they use for the first PCR. Did DNA concentration was quantify after extraction? What is authors’ meaning with "independently performed" second primer pair? According the protocol developed by Hellgre et al. 2004 it have to be use 1 microliter of the first PCR product to run the second PCR in order to increase the sensitivity of the method. As approval, it seems that the authors manage to obtain only 16 sequences out of 30 infected pigeon (L128-129).
>>>In the original manuscript, we omitted some details relating to the molecular methods. We have added some explanations to the text. Blend Taq-Plus is a set of a Taq DNA polymerase, a special PCR buffer adjusted to the aforementioned polymerase, and dNTPs for its optimal PCR reaction. The original protocol is shown in the instruction provided by TOYOBO as an example, and following it we calculated the rate of each constituent for 20µl total PCR reaction solution. We believe these methodology is easily understood by readers by our description. We did not conduct the nested PCR described by Hellgren et al. (2004), but used “independently” the same primer sets written by them. Probably due to high infection intensity of our materials, the nested PCR was not necessary. To avoid readers’ confusion, we have added an explanation related the sentence as follows; ‘Another primer pair of the forward HaemF (5’- ATG GTG CTT TCG ATA TAT GCA TG -3’) and the reverse HaemR2 (5’- GCA TTA TCT GGA TGT GAT AAT GGT -3’), usually used as a second-round primer pair of nested PCR using products of the aforementioned PCR [52], was independently performed [53].’ Concerning to another question given by the reviewer, since we did not conduct the nested PCR, we have not added volume of PCR products for the second-round PCR. Since we examined farmed pigeons, not feral ones, we thought that arbitrarily selected blood samples are enough to know the genetic background of parasites at the time of sample collection. Based on this idea, we collected blood samples for PCR from 14 pigeons at that time.
Third, it is not clear why cloning procedure was applied to some of the samples (L302-307). There are also missing references for the universal primers M13. Cloning is usually applied for sequencing of co-infections in the same individual host (see Pérez-Tris & Bensch, 2005; doi:10.1017/S003118200500733X). Actually, it is not clear how the authors identify co-infections if there are any?
>>>By checking chromograms of direct sequencing, we could know possible cases of co-infections in the same individuals. The same approach has been conducted by Pérez-Tris & Bensch (2005) and other researchers. For clarifying the infection status of H. columbae in our pigeons, we have added new Table 3 (old Table 3 becomes new Table 4).
The performed phylogenetic analyses (Fig. 2) did not bring new information and can be desecrated. It is also not clear how many cytb sequences are included in the phylogeny and why there is no outgroup to root the tree? In my opinion including identical sequences are not giving good results and can be excluded from the tree. Much representative perhaps would be to have a table with cytb lineages that were found, morphospecies, parasitaemia and location. As there is no taxonomical or systematical question in the manuscript, I found the phylogeny quite unnecessary.
>>>We have modified Fig. 2 (phylogenetic tree based on cytb sequences), and added an additional table (new Table 3). The phylogenetic trees based on cytb nucleotide sequences were constructed step by step, and the primary tree of Haemoproteus spp. (80 sequences without our sample sequences) was constructed with Leucocytozoon schoutedeni (DQ676823; genetic lineage GALLUS06) as an outgroup. Second, we constructed a phylogenetic tree of Haemoproteus (Haemoproteus) spp. with Hemoproteus (Parahaemoproteus) spp. as an out group. Third the phylogenetic tree shown as Fig. 2) was constructed. To increase the resolution, we choose the third phylogenetic tree for the representative in the manuscript. Topology of analyzed sequences was always checked to be consistent. This phylogenetic tree is not new, but this figure can provide the background knowledge of phylogenetic relationships of genetic lineages based on cytb for readers. Lineage ‘CXNEA02’ found in mosquitoes Culex neaveai from Cameroon (Njabo et al., 2011) was for the first time from avian hosts (pigeons).
The discussion and conclusions must be entirely revised. In its current version the discussion is too generous and not focused on the main results of the study, i.e. hematogram comparison with infection. Suggestions for superinfection based on the immature to mature gametocytes ratios is too speculative and vague. As the authors did not control the infection in experimental conditions it is hard to say if the young gametocytes infecting erythrocytes are due to reinfection by vector bite or recurrences of the meronts if the hard tissues of the host. I think it will be much more interesting to discuss the results in terms physiological and immunological manner as well.
>>>Since this study was conducted on farmed pigeons with natural infection, there are limitations in clarity, but description of the natural infection is necessary for the future experimental infection study. We suggest and emphasize the importance to analyze the premunition status induced by H. columbae in pigeons in future.
Some of the same and other comments and suggestions I implemented in the attached PDF file.
>>>Thank you for kind review, and we have checked them individually as explained below.
- L12, L35, L42, L50: As mentioned above, we apply the term ‘malaria’ to the disease caused not only by Plasmodium, but also those caused by Hepatocystis and Haemoproteus spp. according to Martinsen, Perkins and Schall (2008) [Ref. 3 in this study].
- L29: We have replaced ‘columvae’ by ‘columbae’. Thank you.
- L56: No change, since ‘hosts’ could not be replaced by ‘birds’. Please compare ‘accidental hosts’ and ‘accidental birds’ for the sentence.
- L100-L101: We have deleted the sentence ‘Microgametocytes, not macrogametocytes, had a clear halo around their nuclei.’
- L102-103: Yes, I agree with your opinion. As shown Figure 1G, which shows a blood smear of the most intensely infected pigeon (assessed by level of parasitemia), multiple infection of the erythorocytes was frequent. To analyze the correlation between levels of parasitemia and multiple infection of the erythrocytes, sample number of the present study is limited. In future, we would like to pursue this issue by increasing sample numbers.
- 2 (the identical sequences can be discarded from the phylogeny.): We have modified Figure 2 to be a compact phylogenetic tree.
- L123: Details are shown in Fig. 2, and this has been explained at L318-319.
- L128: As explained above, and the response to Reviewer#2 (Lines 129 and 311), we had conducted two independent sets of PCR using different primer pairs, not a nested PCR. By a primer combination of HaemNF1 and HaemNR3, we obtained 571-bp long amplicons, and by that of HaemF and HaemR2, 478-bp long amplicons.
- L133: The word ‘promient’ is replaced by ‘dominant’.
- L133, L135: Please see a reference Pérez-Tris & Bensch (2005). The same approach had been done in this study.
- L134: We have newly prepared ‘new Table 3’ (old Table 3 becomes new Table 4), which will support readers’ understanding of this sentence.
- L139: This claim seems not to be right. By the previous phrase, ‘the haemosporidians’ here is limited to ‘Hamoproteus’ To avoid repeated use of a word, we write the sentence like this.
- L212-L217 (This is not connected with the results! It can be deleted. Much more interesting is to compare the haematological parameters you obtain with other studies and to discuss how is that connected with infection of columbae): We have different opinion. We analyze farmed domestic pigeons, not feral domestic pigeons, then their productivity and management are important issues for us.
- L226-L228 (this is too speculative and can not be proven by your results! Furthermore, the place for hypothesis is in introduction. This is not hypothesis that you can prove or not by your results!): This sentence attempts to explain the differences between evident anemia observed in the present study and not-evident anemia in experimental infections. We have deleted this sentence because the same idea is mentioned in the previous sentence.
- L263: We followed the standard method, and commercially available Giemsa’s solution was used. We have added the company name for the Giemsa’s solution company.
- L279: Thank you for your indication. We have added details of hematological analyses that we conducted.
- L283: Please see L143-L146, where abbreviations are explained.
- L291, L295-L297: Please see our explanations mentioned above.
- L307: Universal M13 forward and reverse primers are commonly used in cloning of DNA sequences, and their information is available in the instruction of the plasmid vector pTA2 (TArget Clone™; TOYOBO) mentioned at L303. We have added some words ‘accord-289 ing to the instructions of the manufacturer’ for their explanation.
- L311: See above.
- L334: See above.
- L339: Premunition, also known as infection-immunity, is a host response that protects against high numbers of parasite and illness without eliminating the infection. In this study, pigeons that were infected with mature and immature columbae gametocytes. Mature and immature gametocytes might indicate repeated or recurrent infections in long-lasting periods. Low premunition is supposed to be the cause of this condition. In this sense, pigeon might have incomplete premunition. To avoid confusion, we have replaced ‘superinfection’ with ‘reinfection’.
Reviewer 2 Report
This is a well written report providing evidence that Haemoproteus columbae infections in farmed domestic pigeons are not entirely irrelevant from the animal health perspective but can result in hypochromic anemia and hypoproteinemia.
I do only have a few minor comments:
Title: Remove the “the” between “From and “Central Java”:
Impact of Subclinical Haemoproteus columbae Infection on Farmed Domestic Pigeons from Central Java (Yogyakarta), Indonesia, with Special Reference to Changes in the Hemogram
Line 137: …that infect pigeons and…
Table 3: in the current version the borders between the columns are not easily recognizable. I recommend to always place the values for arithmetic mean and standard deviation (in brackets) below the range of values.
Line 166: …island of Tenerife…
Line 191: … In accidental hosts… (remove “the”)
Line 424: the second author´s name is “Harl”
Lines 129 and 311: I did not understand why a part of the sequences had a 571 bp length. Could you please explain the significance of these approx. 100 additional bps?
Author Response
Thank you a lot for your kind review with invaluable comments and suggestions. Based on your comments, we have revised the manuscript.
A point-by-point responses to your comments are as follows;
This is a well written report providing evidence that Haemoproteus columbae infections in farmed domestic pigeons are not entirely irrelevant from the animal health perspective but can result in hypochromic anemia and hypoproteinemia.
I do only have a few minor comments:
- Title: Remove the “the” between “From and “Central Java”: ‘Impact of Subclinical Haemoproteus columbae Infection on Farmed Domestic Pigeons from Central Java (Yogyakarta), Indonesia, with Special Reference to Changes in the Hemogram’.
>>>We have removed ‘the’ prior to ‘Central Java’, and changed the title as indicated by the reviewer.
- Line 137: …that infect pigeons and…
>>>Thank you for indicating our typographical error which we had overlooked. We have corrected it as indicated.
- Table 3: in the current version the borders between the columns are not easily recognizable. I recommend to always place the values for arithmetic mean and standard deviation (in brackets) below the range of values.
>>>We have modified Table 3, in which arithmetic mean and standard deviation (in brackets) below the range of values.
- Line 166: …island of Tenerife…
>>>Thank you for your kind indication, and we have corrected it.
- Line 191: … In accidental hosts… (remove “the”)
>>>We have remove ‘the’ prior to ‘accidental hosts’ as indicated.
- Line 424: the second author´s name is “Harl”
>>>Thank you, and we have corrected the name of the second author’s name.
- Lines 129 and 311: I did not understand why a part of the sequences had a 571 bp length. Could you please explain the significance of these approx. 100 additional bps?
>>>We used two primer sets to amplify partial cytb sequences, i.e. primer combination of HaemNF1 and HaemNR3, and HaemF and HaemR2. By the former primer pair, we obtained 571-bp long amplicons, and by the latter 478-bp long amplicons. Usually these two primer combinations are used in a nested PCR as the first round primer pair and second one, but in this study we used these primer combinations separately. PCR using he former primer pair was successful for obtaining seven sequences.
Round 2
Reviewer 1 Report
This is a second revision of the manuscript. Generally, the authors appropriately addressed most of my concerns and suggestions from the previous version, but there are several methodological issues left, that requires attention again. From the answers of the authors, I understood that they select 14 of the 30 infected pigeons to be screened by PCR and obtain sequences for the phylogenetic analyzes. That must be clearly explained in the materials and methods section! Why this samples were selected, on what basis and how that will affect the results? In its second version, I was again misunderstanding that the authors didn’t manage to amplify and sequence successfully around 50% of the positive samples! I would ask the authors to explain why they changed the developed by Hellgren et al. 2004 high sensitive nested PCR method and perform two independent PCR with the nested primes? Furthermore, it have to be clearly explained in the materials and methods (not only in answers to the reviewer) section how the authors identify the co-infections! It still not clarified how much of the sample was used as a template for the PCR amplification!
The term “pigeon malaria” is not used in any of the cited references in the manuscript! I found it only in quite old study by Kander (1941) (Kadner, C. G. (1941). Pigeon Malaria in California. Science (Washington), 93(2412), which is not cited in the manuscript. I will again insist to use the widely accepted by researchers in the field "haemoproteosis", even more it is appropriate term according to the mentioned references 2, 5, 6! As the phylogeny of order Haemosporida is still under debate (doi: 10.1098/rsos.171780) I would recommend to stick with the traditional epidemiological and malariological meaning of the term "malaria" (please see doi: 10.1016/j.pt.2005.06.005).
I will again recommend to remove the identical sequences from the phylogenetic analyzes! The current phylogenetic tree is overloaded with information for GenBank accession numbers, location, sample id etc. In my opinion it can remain only species and MalAvi abbreviation id. Moreover, it is misleading when for instance it is shown LC605998–LC606000 if this is single lineage or there are concatenated lineages. I will ask again the authors to specify either in the figure caption or in the methods how many lineages are used for building the tree!
Some of the same and other comments and suggestions I implemented in the attached PDF file.

Author Response
Reviewer#1
Thank you a lot for your kind review with invaluable comments and suggestions. Based on your comments, we have revised the manuscript.
- This is a second revision of the manuscript. Generally, the authors appropriately addressed most of my concerns and suggestions from the previous version, but there are several methodological issues left, that requires attention again. From the answers of the authors, I understood that they select 14 of the 30 infected pigeons to be screened by PCR and obtain sequences for the phylogenetic analyzes. That must be clearly explained in the materials and methods section! Why this samples were selected, on what basis and how that will affect the results? In its second version, I was again misunderstanding that the authors didn’t manage to amplify and sequence successfully around 50% of the positive samples! I would ask the authors to explain why they changed the developed by Hellgren et al. 2004 high sensitive nested PCR method and perform two independent PCR with the nested primes? Furthermore, it have to be clearly explained in the materials and methods (not only in answers to the reviewer) section how the authors identify the co-infections! It still not clarified how much of the sample was used as a template for the PCR amplification!
>>>We examined farmed pigeons, not feral pigeons. In such cases, most of researchers choose arbitrarily selected samples, not all samples, when sample numbers are not few. We supposed at that time covering of appropriate numbers of samples is enough to know the dominant lineages of pathogens. We believe that most researchers could have the same idea by using the words ‘arbitrarily chosen samples’. We have added ‘one µl of template DNA’ in the sentence for PCR reaction according to the instruction provided by the company (Toyobo), which had been written in the original manuscript. Why always we have to use the nested PCR? Hellgren et al. (2004) used these two sets of PCR primers for the nest PCR, but we could use them in independent PCR reactions. They amplified different lengths of sequences, then efficiency of sequencing could be different. We have described that, in the original and revised manuscript, by cloning we could identify the co-infection. We believe that readers could understand it, since this is a common approach to detect the co-infection without special explanation.
- The term “pigeon malaria” is not used in any of the cited references in the manuscript! I found it only in quite old study by Kander (1941) (Kadner, C. G. (1941). Pigeon Malaria in California. Science (Washington), 93(2412), which is not cited in the manuscript. I will again insist to use the widely accepted by researchers in the field "haemoproteosis", even more it is appropriate term according to the mentioned references 2, 5, 6! As the phylogeny of order Haemosporida is still under debate (doi: 10.1098/rsos.171780) I would recommend to stick with the traditional epidemiological and malariological meaning of the term "malaria" (please see doi: 10.1016/j.pt.2005.06.005).
>>>We have delteted ‘pigeon malaria’ and used ‘pigeon haemoproteosis’ in the revised manuscript.
- I will again recommend to remove the identical sequences from the phylogenetic analyzes! The current phylogenetic tree is overloaded with information for GenBank accession numbers, location, sample id etc. In my opinion it can remain only species and MalAvi abbreviation id. Moreover, it is misleading when for instance it is shown LC605998–LC606000 if this is single lineage or there are concatenated lineages. I will ask again the authors to specify either in the figure caption or in the methods how many lineages are used for building the tree!
>>>We have deleted the phylogenetic tree (Fig. 2) from the main text, and transferred it to Supplementary Figure in the revised manuscript.
Some of the same and other comments and suggestions I implemented in the attached PDF file. We could not understand about your concern about ’LC605998–LC606000’.
A point-by-point responses to your comments are as follows;
L11: I think "pigeon haemoproteosis" will better fit the current knowledge on the epidemiology, life history and evolution of this parasites!
>>>According to your advice, we have replaced ‘pigeon malaria’ by ‘pigeon haemoproteosis’ in the revised manuscript.
L28: The study was not designed in a manner that the productivity of the pigeons can be measured and analyzed. It is also no clear what the authors mean with "productivity", production of meat, eggs, grown pigeons or something else?
>>>We have deleted the word ‘productivity’ from this sentence.
L28-L29: Once again, for me it is unclear how this statement can be main conclusion of the study that is not investigating number of vector bites and did not follow the dynamics of parasitaemia for certain period of time? I think it is too speculative to have such conclusion in the abstract that will mislead the reader.
>>>We have deleted the phrase which the reviewer indicted to be too speculative.
L31: Please be consistent and use either "haplotype" or "lineage"!
>>>According to the reviewer’s comment, we replace all ‘haplotype’ by ‘lineage’ in the revised manuscript.
L35-L36: Haemoproteus spp. cause hemoproteosis and Leucocytozoon spp. leukocytozoonosis! Although for Haemoproteus spp. there is a debate as to whether it is malaria or not, then for Leicocytozoon there is no doubt that it is not malaria.
>>>We have modified the sentence as follows: ‘Avian haemosporidian infection is caused mainly by the genera Plasmodium Mar-chiafava et Celli, 1885, Haemoproteus Kruse, 1890, or Leucocytozoon Sambon, 1908 (Apicomplexa: Haemosporidia).’
L41: The term is not used in any of the cited references! I found it only in quite old study by Kander (1941) (Kadner, C. G. (1941). Pigeon Malaria in California. Science (Washington), 93(2412), which is not cited in the manuscript. I will again insist to use the widely accepted by researchers in the field "haemoproteosis" even thought it is appropriate term according to the mentioned references 2, 5, 6! As the phylogeny of order Haemosporida is still under debate (doi: 10.1098/rsos.171780) I would recommend to stick with the traditional epidemiological and malariological meaning of the term "malaria" (please see doi: 10.1016/j.pt.2005.06.005).
>>>We understand your opinion, and we have not used ‘pigeon malaria’ in this revised manuscript.
L48: Please reconsider using of this term.
>>>We have replaced ‘pigeon malaria’ by ‘pigeon haemoproteosis’.
L55: columbid?
>>>Yes. Thank you for your kind indication. We have corrected it.
L57: If the authors insist that this is "malaria" why they are not consistent here?
>>>The terms ‘malaria’ and ‘haemoproteosis’ have partially overlapped tems, not identical. Even, the term ‘haemoproteosis’ has different implications by cases, as you know.
L67, L70, Table 1: Parasitemia?
>>>We have replaced ‘intensity’ by ‘parasitemia’ or ‘levels of parasitemis’.
L99, L100: pigment granules?
>>>We have replaced ‘cytoplasmic granules’ by ‘pigment granules’.
L115: I would recommend to show with an arrows the main morphological features that were described on L 92-103
>>>We have added ‘Nuclei (arrowheads) and pigment granules (pg) of H. columbae are shown in the photographs B and E. Arrow in the photograph G indicates an erythrocyte with two gametocytes.’ to the legend for Fig. 1, and indicated them in the figure. Thank you for your advice.
Table 3: Please check this again it seems that these are only the infected pigeons! Replace with co-infected.
>>>Yes, they were infected and examined molecular-genetically. We have added ‘molecular genetically’ to the place you indicated. We have replaced ‘mixed infected’ by ‘co-infected’. We have modified two roman words which should be in italic; ‘cytb’ and ‘H. columbae’. We have deleted unnecessary ‘LC605998–LC606001’ in the table.
Table 4: n.d.3?
>>>Yes, your indication is right, and we have corrected.
L180: I would recommend to use just parasitemia and to avoid using of intensity.
>>>We have replaced ‘infection intensity’ by ‘parasitemia levels’.
L190: Please check again this statement. How it is possible that the acute phase is before prepatent period? From the statement here it appears that prepatent period is 22 - 37 days post infection and acut phase is 9 - 22 days post infection! According the referred study by Cepeda et al. (32) it should be 19-20 days post infection prepatent period and 22-25 days post infection acute stage, afterwords, the chronic stage should start after 29-30 post infection.
>>>You misunderstood this sentence: The prepatent period (merogony) is followed by the acute phase (gametogony). Merogony in the repatent period, and gametogony in the patent period. We did not use the term ‘postinfection’ after ‘days’. Then, readers could understand the period between 22 and 37 days means the duration. Similarly, the duration of the acute phase ranges between nine and 20 days. It is important to know how long different phases of the disease continue for our study.
L216: It seems you are familiar with the appropriate terminology.
>>>We always choose the best term for the sentence.
L274-276: This should be moved to the 4.3. DNA extraction, amplification, and sequencing section. Or at least must be repeated.
>>>We keep this sentence here, and repeat some information in the section 4.3 as follows: ‘As mentioned above, blood from 14 arbitrarily selected pigeons (Mlati [n=4], Ngemplak [n=5], Kalasan [n=3], and Sedayu [n=2]) were used for DNA extraction.’
L298: microhematocrit?
>>>Yes, you are right, and we misspelled it. We have corrected it.
L314, L319: Please specify how many samples?
>>>As mentioned above, we have specified the number of samples.
L324: Would you explain why you changed the developed by Hellgren et al. 2004 high sensitive nested PCR method and perform two independent PCR with the nested primes?
>>>Due to the high level of parasitemia in our samples, high sensitive PCR was not necessary. It is known that PCR using different primer pairs might detect different dominant sequences, particularly in Leucocytozoon infection in birds. In our past study, PCR using different primer pair detected different dominant sequences of Leucocytozoon spp. The same findings have been published by at least two articles (details we forgot). We tried to test this possibility. PCR amplification and sequencing using a primer pair of HaemF and HaemR2 were successful for all 14 samples, whereas PCR amplification using a primer pair of HaemNF1 and HaemNR3 was successful similarly for all 14 samples, but sequencing was successful partly, and seven sequences were obtained. There were no discrepancies between sequences amplified by different primer pairs.
L363: Pigeon haemoproteosis?
>>>We have replaced ‘pigeon malaria’ by ‘Pigeon haemproteosis’.
L363: Mostly in the tropics!
>>>We have replaced ‘worldwide’ by ‘mostly in the tropics and subtropics’.
L372-L374: This can be deleted or replaced as it is not a direct outcome of the results.
>>>We have deleted this sentence as you recommended.
